# Unfolded Protein Response (UPR) in Survival, Dormancy, Immunosuppression, Metastasis, and Treatments of Cancer Cells

**DOI:** 10.3390/ijms20102518

**Published:** 2019-05-22

**Authors:** Sheng-Kai Hsu, Chien-Chih Chiu, Hans-Uwe Dahms, Chon-Kit Chou, Chih-Mei Cheng, Wen-Tsan Chang, Kai-Chun Cheng, Hui-Min David Wang, I-Ling Lin

**Affiliations:** 1Department of Biotechnology, Kaohsiung Medical University, Kaohsiung 807, Taiwan; b043100050@gmail.com (S.-K.H.); cchiu@kmu.edu.tw (C.-C.C.); 2Department of Medical Laboratory Science and Biotechnology, Kaohsiung Medical University, Kaohsiung 807, Taiwan; 3Department of Biological Sciences, National Sun Yat-sen University, Kaohsiung 804, Taiwan; 4Center for Stem Cell Research, Kaohsiung Medical University, Kaohsiung 807, Taiwan; 5Department of Medical Research, Kaohsiung Medical University Hospital, Kaohsiung 807, Taiwan; chmech@kmu.edu.tw; 6The Graduate Institute of Medicine, Kaohsiung Medical University, Kaohsiung 807, Taiwan; 7Department of Biomedical Science and Environmental Biology, Kaohsiung Medical University, Kaohsiung 807, Taiwan; hansd@kmu.edu.tw; 8Department of Marine Biotechnology and Resources, National Sun Yat-sen University, Kaohsiung 804, Taiwan; 9State Key Laboratory of Quality Research in Chinese Medicine, Institute of Chinese Medical Sciences, University of Macau, Macau, China; fatchou1988@hotmail.com; 10Division of General and Digestive Surgery, Department of Surgery, Kaohsiung Medical University Hospital, Kaohsiung 807, Taiwan; wtchang@kmu.edu.tw; 11Department of Surgery, School of Medicine, College of Medicine, Kaohsiung Medical University, Kaohsiung 807, Taiwan; 12Department of Ophthalmology, Kaohsiung Municipal Hsiaokang Hospital, Kaohsiung 812, Taiwan; pington64@gmail.com; 13Department of Ophthalmology, Kaohsiung Medical University Hospital, Kaohsiung 807, Taiwan; 14Graduate Institute of Biomedical Engineering, National Chung Hsing University, Taichung 402, Taiwan; davidw@dragon.nchu.edu.tw

**Keywords:** endoplasmic reticulum (ER), unfolded protein response (UPR), PERK, IRE-1, ATF6, cancer

## Abstract

The endoplasmic reticulum (ER) has diverse functions, and especially misfolded protein modification is in the focus of this review paper. With a highly regulatory mechanism, called unfolded protein response (UPR), it protects cells from the accumulation of misfolded proteins. Nevertheless, not only does UPR modify improper proteins, but it also degrades proteins that are unable to recover. Three pathways of UPR, namely PERK, IRE-1, and ATF6, have a significant role in regulating stress-induced physiological responses in cells. The dysregulated UPR may be involved in diseases, such as atherosclerosis, heart diseases, amyotrophic lateral sclerosis (ALS), and cancer. Here, we discuss the relation between UPR and cancer, considering several aspects including survival, dormancy, immunosuppression, angiogenesis, and metastasis of cancer cells. Although several moderate adversities can subject cancer cells to a hostile environment, UPR can ensure their survival. Excessive unfavorable conditions, such as overloading with misfolded proteins and nutrient deprivation, tend to trigger cancer cell death signaling. Regarding dormancy and immunosuppression, cancer cells can survive chemotherapies and acquire drug resistance through dormancy and immunosuppression. Cancer cells can also regulate the downstream of UPR to modulate angiogenesis and promote metastasis. In the end, regulating UPR through different molecular mechanisms may provide promising anticancer treatment options by suppressing cancer proliferation and progression.

## 1. Introduction

### 1.1. The Main Functions of the Endoplasmic Reticulum in Eukaryotic Cells

The endoplasmic reticulum (ER) is a significant component of the endomembrane system in eukaryotic cells. It has diverse functions, such as in the synthesis of fats, metabolism of glucose, detoxification, Ca^2+^ homeostasis, and protein modification [1,2]. ER is a highly regulated system that modifies dysfunctional proteins and prevents them from secretion. Several adverse conditions, such as hypodermic hypoxia, accumulation of reactive oxygen species, lack of ATP, nutrient deprivation, and mutations in specific proteins may lead to the accumulation of misfolded proteins in the ER. To keep the cell working normally and not harming other cells, the ER tends to stimulate an unfolded protein response (UPR) [3]. ER-triggered UPR affects the problem of misfolded and unfolded proteins, mainly through four mechanisms. Initially, it will immediately shut down the production of proteins to protect from the synthesis of more misfolded proteins. In this first line of response, UPR induces translation attenuation through modulating elF2α (eukaryotic Initiation Factor 2α) and cell cycle arrest by the PERK pathway (protein kinase RNA-activated (PKR)-like ER kinase) [4]. Besides, UPR can also induce the expression of chaperones, such as Grp78/Bip, HSP40 ERdj3/DNAJB11, and HSP70 [5], which can repair misfolded proteins. Moreover, if misfolded and unfolded proteins are overloaded, the ER is likely to discard those misfolded proteins and even degrade them by the proteasome. However, sometimes, if the condition is out of control, UPR may finally trigger cell death signals.

### 1.2. The Roles of UPR in Coping with Misfolded Proteins and Its Downstream Proteins

The function of UPR is to maintain homeostasis by strengthening the ability of protein folding in the ER via chaperones. Furthermore, it can also promote ERAD (ER-associated degradation) pathways or chaperones to cope with abnormal proteins and even suppress other mRNA translation to attenuate more protein synthesis [6]. ERAD also ensures that proper and normally folded proteins can be translocated to their normal destination [1]. Thus, misfolded and unassembled proteins will be tackled by ERAD-designated cellular pathways, including ubiquitination by ubiquitin and degradation by the proteasome, which decreases the accumulation of abnormal and unsalvageable proteins in the ER. Besides, the mechanism of ERAD is a four-continuous-step process: firstly, a misfolded glycoprotein is recognized, and then this will be retro-translocated from the ER to the cytoplasm. Immediately following this, a misfolded protein will be ubiquitinated by E3 ubiquitin ligases and subsequently degraded by the proteasome [7].

UPR protects cells in two respects. In general, it plays a crucial role in maintaining cell survival by ensuring the quality of secretory proteins. However, overloaded stressors, such as persistent hypoxia and nutrient deprivation of cells or even increased the accumulation of reactive oxygen species, may activate cell death signals, not cell adaptation [8]. Normally, Grp78/Bip binds closely to the N-terminal side of three transmembrane proteins in the lumen of ER, including IRE-1 (Inositol-requiring transmembrane kinase/endoribonuclease), PERK (Protein kinase RNA-activated (PKR)-like ER kinase) and ATF6 (Activating transcription factor 6), to prevent their activation and block the downstream signaling of UPR [9,10]. However, in the presence of excessive unfolded proteins, Grp78 moves from its interaction with PERK, ATF6 and IRE-1 to act as chaperones. This allows activation of ATF6 and PERK and IRE-1 to dimerize and become active [9].

In mammalian cells, there are three integral ER membrane proteins: (1) IRE-1-XBP1; (2) PERK-ATF4-CHOP; and (3) ATF6 [11] (Figure 1). These proteins serve as stress sensors to regulate the production of bZIP domain-containing transcription factors, including XBP1s, ATF4, and cleaved ATF6, which upregulate genes involved in ER functions [12]. For example, XBP1s targets for the genes involved in chaperones synthesis, lipid synthesis, and ERAD. ATF4 is responsible for the genes associated with cell viability, such as CHOP (C/EBP homologous protein), growth arrest, DNA damage-inducible protein GADD34, amino acid metabolism, and redox homeostasis [13]. Cleaved ATF6 targets the genes related to chaperones, such as Grp78/Bip, lipid synthesis, and apoptosis. During the stress response, the release of Grp78 from the lumen of ER activates downstream signaling to convey signals of UPR to the specific genes (Figure 1). IRE-1-XBP1 pathway has several physiological functions, such as inducing ERAD, modulating the fatty acid synthesis and regulating protein secretion [13]. Mammalian IRE-1 has two isoforms, including IRE-1α and IRE-1β. IRE-1α is a transmembrane protein kinase, ubiquitously expressed on the ER membrane within each cell.

However, IRE-1β is merely expressed in the epithelial cells of the gastrointestinal tract. IRE-1α has two enzymatic domains: one is the Ser/Thr kinase domain, and the other is the site-specific endoribonuclease (RNase), which autoregulates its mRNA for UPR [15,16]. With the accumulation of misfolded proteins, Grp78/Bip removes from its interaction with IRE-1. Free Grp78/Bip will bind with unfolded proteins and provides conformational changes in IRE-1, which causes IRE-1 to dimerize and autophosphorylate. Then, the activated RNase can remove 26-nt introns from XBP1 (X-box-binding protein 1) and transforms mRNA into the XBP1s forms, serving as transcription factors [17]. Transcription factors can translocate to the nucleus and bind to UPRE (unfolded protein response element), which induces the expression of genes involved in ER chaperone (Grp78), ERAD, and lipogenesis pathway components.

The IRE-1 pathway also suppresses the expression of other mRNAs to reduce excessive unfolded proteins. It can also lead to microRNA degradation by RIDD (regulating an IRE1-dependent decay), including miR-17, miR-34a, miR96, and miR-125b [18]. PERK serves as another ER transmembrane kinase. When Bip is released from its luminal binding domain, PERK is activated by autophosphorylation. Then, active PERK can phosphorylate the elF2α factor, which has a significant role in the initiation of protein translation in eukaryotic cells, to suppress the process of translation during protein synthesis [19]. However, a transcription factor, known as ATF4 (Activating transcription factor 4), is related to the expression of chaperones.

Furthermore, it also activates CHOP to trigger cell death signals. CHOP, also known as DNA damage-inducible transcript 3, is a pro-apoptotic transcription factor [20]. During ER stress, it can induce Ero1 (ER oxidoreductin 1), causing calcium release from the ER into the cytoplasm, and resulting in apoptosis [21]. It also induces apoptosis through activating growth arrest and DNA damage-inducible protein GADD34. GADD34 targets α isoform of protein phosphatase (PP1), and hence promotes dephosphorylation of elF2α to enhance translation [11,22].

ATF6 as the other transmembrane protein is a basic leucine zipper transcription factor. Once Grp78/Bip dissociates, it gets transported to the Golgi body and is activated by RIP (Regulated intramembrane proteolysis), including S1P (Site-1 protease) and S2P (Site-2 protease), to form an active transcription factor that translocates to the nucleus. The active ATF6 induces the expression of CHOP, chaperones, and ERAD components [23]. Furthermore, ATF6 can regulate lipid metabolism, which increases ApoB-100 protein levels during glucose deprivation. Thus, it can be concluded that, during ER stress, three transmembrane proteins of UPR may induce the expression of CHOP, which in the first place leads to apoptotic signaling [24] (Figure 1).

## 2. UPR and Cell Survival

### 2.1. UPR in Cell Survival

It is significantly revealed that ER stress and UPR activation play vital roles in the development of cancer [25]. UPR signaling not only attenuates mRNA translation for inhibiting over-produced misfolded proteins but also upregulates the expression of chaperones to repair misfolded proteins [26]. However, overloaded ER stress, such as the accumulation of misfolded proteins and the inactivation of Grp78/Bip, may induce cell death signaling. By contrast, cancer cells manifest different responses compared to normal cells. Interestingly, during unfavorable conditions, such as hypoxia, oxidative stress, nutrient deprivation, and low pH, cancer cells tend to modify their ER-resident proteins and chaperones to increase cell viability. Treating tumor-bearing mice (mouse TC-1, cervical cancer cells) with ER stress inducers, such as thapsigargin, a selective inhibitor of SERCA (sarcoplasmic-endoplasmic reticulum calcium ATPase), causes an increase of tumor growth [27].

When using UPR inhibitors, such as 4-PBA (4-Phenylbutyric acid) or TUD-CA (Tauroursodeoxycholic acid), tumor progression and metastasis are delayed [28]. Therefore, with a moderate increase in ER stress, integral ER membrane proteins may play a crucial role in cell survival. In another case, MEFs (mouse embryonic fibroblasts) treated with persistent low-grade pharmacological ER stress resist ER stress for cell survival owing to higher mRNA stability of pro-survival Bip and decreasing mRNA stability of CHOP [29]. IRE-1 can trigger both adaptive and death pathways by its endoribonuclease activity. Furthermore, RIDD can also initiate adaptive and death pathways. In an adaptive pathway, activated IRE-1 can promote expression of XBP-1 and indirectly induce ERAD, which aids cell survival [30]. For example, XBP-1KO/Eμ-TCL1 mice develop leukemia significantly more slowly than XBP-1WT/Eμ-TCL1 mice [31].

Besides, hypoxia and nutrient deprivation can induce breast cancer cell line growth viability through XBP1 splicing [32]. Another ER-associated transmembrane protein PERK can increase cancer cell viability during adversities, such as hypoxia, ATP shortage, and nutrient deficiency. Although not all protein synthesis is eIF2α-dependent, activated PERK could phosphorylate eIF2α to inhibit eIF2α-dependent protein synthesis. For example, according to Koumenis results, PERK-induced elF2α and ATF4 protects tumor cells through overcoming hypoxia [33]. Once eIF2α is blocked, ATF4 is induced to up-regulate the genes with roles in protein synthesis and antioxidant responses for survival. It was shown that some tumor cells, which are PERK^−/−^ under hypoxia, have lower viability and lost their ability of angiogenesis [34].

When the PERK gene in mouse embryonic stem cells is mutated, the cells with homozygous mutation express 10-fold less *PERK* mRNA than wild-type cells, which leads to a decreased level of phosphorylation of eIF2α [35]. Furthermore, it is reported that ATF4-induced miR-211 decreases the expression of CHOP due to hypermethylation on its promoter [36]. It is reported that *CHOP* knockout leads to lung lesion in an immunocompetent K-RasG12V mutation-driven murine model of lung cancer [32]. Furthermore, it is reported that up-regulation of Grp78 on cancer cell plasma membranes leads to cell survival and induces MAPK (mitogen-activated protein kinase) and PI3/Akt (protein kinase BPKB) pathways [37]. On the other hand, PERK can not only activate ATF4 to strengthen cell survival but also induce the activation of Nrf2 (Nuclear factor 2), a transcription factor, to inactivate CHOP, which blocks cell death signaling [38] (Figure 2a). This leads to the conclusion that UPR is beneficial for cancer cells. Therefore, using UPR inhibition, although it is still difficult to completely shut down cancer cell growth, it can slow down progression and metastasis.

### 2.2. UPR in Cell Death

However, UPR is a two-edged sword, playing a role in cell survival as well as cell death. Activated ATF4 can promote the expression of CHOP/GADD153 (transcription factor for apoptotic protein, Bim) and subsequently induces Bim and inhibits Bcl-2, Bcl-XL and Mcl-1 (anti-apoptotic proteins) [39,40]. Furthermore, CHOP can also be activated by ATF6 and sXBP1. Besides ATF4, IRE-1 can be functional as a cell death trigger. IRE-1α, one of the IRE-1 isoforms, can recruit TRAF2 to ASK1 and its downstream target JNK/MAPK8/SAPK1 (c-Jun N-terminal kinase 1) under sustained engagement. In summary, the IRE-1-mediated JNK pathway could promote both apoptotic an non-apoptotic cell death [41] (Figure 2b).

## 3. UPR and Tumor Dormancy

### 3.1. UPR-Induced Dormancy in Cancer Metastasis

Cancer dormancy can roughly refer to two different types: One is tumor mass dormancy, and the other is cellular dormancy [43]. Tumor mass dormancy means that tumor cells usually divide, but their mass is limited owing to deficient blood supply or active immune effects [43]. Cellular dormancy can be defined as cancer cells escaping from the mitotic cell cycle. They stop dividing and survive in a quiescent state where cells are arrested at G0–G1 in the cell cycle to wait for an appropriate microenvironment [43,44]. Two significant components of dormancy are survival and cell cycle G0–G1 growth arrest [33]. Furthermore, dormant cells are considered to be present in the earliest stage of tumor progression [45].

However, the dormant cancer cells are difficult to be detected and to be treated. Several studies supported that DCTs (dissemination tumor cells) enter the dormant state by establishing unstable interactions with the ECM (extracellular matrix) [44]. There are two possible mechanisms leading to cancer cell dormancy: One is a proliferative arrest that an individual cancer cell fails to proliferate. The other is the balance between the cytotoxicity of cytotoxic T lymphocyte and angiogenesis of cancer cells [44]. If the activity of the immune response is prevailing, cancer cells will be eliminated via cytotoxic reaction or apoptosis. In contrast, if angiogenesis is increasing, cancer cells will proliferate. Therefore, when the immune response and the angiogenesis strike a balance, cancer cells tend to be dormant. Furthermore, cancer cells can make use of dormancy to overcome hostile microenvironments, such as hypoxia, lack of nutrients, and chemotherapies. Once these unfavorable conditions are ruled out, cancer cells tend to proliferate persistently.

However, in former studies, it is manifested that the wrestling between MAPK p38 and ERK (Extracellular regulated protein kinases) plays a crucial role in cell dormancy [46]. When tumor cells degrade ECM and vascular walls to disseminate, there are three possible outcomes: Firstly, with proliferative MAPK p38 with decreasing ERK signals, tumor cells degenerate without metastasis during dissemination. Secondly, increasing ERK and MAPK p38 leads to metastasis. Thirdly, decreasing ERK with increasing MAPK p38 brings about metastatic cell dormancy [33,46] (Figure 3a). Moreover, dormancy plays a significant role in both primary tumor proliferation and metastasis. In the past, it was shown that metastasis is a multistep process, involving migration, intravasation, extravasation, and colonization. However, in recent studies, metastasis dissemination can even be detected at an early stage, especially in bone marrow and lung. There are two articles supportive of the phenomena: Firstly, a HER2-driven mouse model of breast cancer showed that progesterone signaling and the promotion of HER2 lead to cancer cell migration at an early stage [47]. Secondly, the downregulation of MAPK p38 and HER2 expression in the same mouse model results in early dissemination [44].

### 3.2. The UPR-Induced Dormancy in Chemoresistance and Cell Survival

Regarding tumor cell dormancy, UPR might play a crucial role. As mentioned above, MAPK p38 not only inhibits ERK but also induces the downstream of UPR, including PERK and IRE-1α. PERK can trigger Nrf2 and phosphorylated-elF2α, which contribute to cell cycle G0–G1 arrest. Besides, IRE-1α can induce XBP1s and subsequently stimulate Grp78/Bip for cell survival. The combination of cell cycle G0–G1 arrest and cell survival leads to tumor cell dormancy and drug resistance [33].

It is widely considered that dormant cancer cells are resistant to chemotherapies owing to their proliferative suppression. Furthermore, it was shown that HEp3, head, and neck squamous carcinoma cells up-regulate all three transmembrane proteins of UPR: ATF-6α, IRE-1α, and PERK [49]. Additionally, under chemotherapies, PERK can induce G0–G1 arrest and promote cancer cell survival [50]. The signaling of PERK-eIF2α contributes to the dramatic arrest of the G0–G1 phase and survival in epidermoid carcinoma cells HEp3 [50]. However, over-expression of PERK may inhibit tumor growth through intruding cell cycle regulators, such as cyclin A, cyclin D1, and cyclin D3 [50]. Furthermore, another investigation reveals that p38 can up-regulate the expression of Grp78 and PERK in dormant HEp3 cells, which induces chemoresistance and cell survival by inhibiting a pro-apoptotic protein, Bax [51]. Actively spliced XBP1 (XBP1s) or ATF6α decrease the numbers of surviving HEp3 cells in a dormant state.

Nevertheless, ATF6α can play another role that induces HEp3 cell survival via the activation of mTOR (Akt-independent mammalian target of rapamycin) signaling, which is regulated by the upregulation of the small GTPase Rheb (Ras homolog enriched in brain). However, the overexpression of Rheb is observed in multiple carcinomas. Thus, inhibiting Rheb and suppressing mTOR signaling provide promising treatments for cancer cells [52,53]. Furthermore, high MAPK p38/ERK ratios lead to an upregulation of p53 and a downregulation of FoxM1 (Forkhead box protein M1) and c-Jun [48] transcription factors (Figure 3b).

## 4. UPR and Immunosuppression in Cancer Cells

As described above, IRE-1α-XBP1 signaling plays a significant role in cancer cell survival. However, the mechanism by which IRE-1α-XBP1 signaling inhibits immune response is not completely understood [32]. In the immunocompetent state, antigenic carrying cells, primarily dendritic cells, provide antigens for T lymphocytes including both CD4+(Th) and CD8+(Tc) lymphocytes to recognize and make them mature. They activate Th by CD40-CD40L interaction (Cluster of differentiation 40) and MHC-II (Major histocompatibility complex II), which upregulates the release of cytokines for cytotoxic lymphocyte activation [54]. Simultaneously, they can also activate Tc via CD80-CD28 interaction and MHC-I (Major histocompatibility complex l) [55].

However, immunosuppression may provide an opportunity for the formation of cancer cells. Furthermore, cancer cells sometimes protect themselves from elimination through UPR-driven immunosuppression. Aggressive cancers can not only recruit immune cells, such as dendritic cells and Tc lymphocytes but also eliminate their original functions, such as cytotoxic effects and antigen presentation [32]. Therefore, if dendritic cells are blocked or inactivated by aggressive cancer cells, T lymphocytes will fail to execute their normal functions, including recognizing and killing cancer cells. Furthermore, under unfavorable conditions, such as hypoxia and lack of nutrients, cancer cells lead to the accumulation of unfolded proteins, which trigger the activation of ER stress and IRE-1α-XBP1 in TDCs (Tumor-associated dendritic cells), and then suppresses their antigen-presenting functions [32]. Besides, a high concentration of ROS (Reactive oxygen species) in TDCs can both promote lipid peroxidation and upregulate the synthesis of 4-HNE (4-Hydroxynonenal), one of the major end products of lipid peroxidation, which induces structural changes of chaperone and ER-resident proteins and triggers IRE1α-XBP1 for suppressing antigen-presentation [32]. Furthermore, immunosuppression is highly associated with metastasis [56]. Ovarian cancer-bearing mice that selectively lack XBP1 genes in dendritic cells showed that ovarian tumors have a slower progression and less metastasis [57]. The effect resulting from inhibition of IRE-1α-XBP1 signaling enhances activated and antigen-experienced T cytotoxic lymphocyte to produce IFN-γ (Interferon gamma), leading to immunocompetence instead of immunosuppression [58].

Moreover, active XBP1 not only up-regulates the XBP1-associated genes involved in ER stress but also promotes triglyceride biosynthesis, which leads to the abnormal accumulation of lipids, such as phosphatidylcholine, the primary phospholipid of the ER membrane [57,59]. Another experiment suggested that increased accumulation of oxidized fatty acids was related to the dysfunction of tumor dendritic cell [60]. In contrast, tumor dendritic cells with inactivated XBP-1 are accessible to strengthen the functions of T lymphocytes, such as maturation of cytotoxic T lymphocytes and memory T lymphocytes, without the aberrant accumulation of intracellular triglycerides [58] (Figure 4).

## 5. UPR and Angiogenesis

### 5.1. Introduction to Angiogenesis

Under stressful conditions, especially hypoxic microenvironments, cancer cells are inclined to take several means to maintain energy balance, including HIF (hypoxia-inducible factor), UPR and macroautophagy. However, angiogenesis is a significant mechanism for tumor cells to maintain their metabolic balance. There are several known factors regulating angiogenesis, including FGF (fibroblast growth factor), PDGF (platelet-derived growth factor), IL-8, and VEGF (vascular endothelial growth factor) [61].

When tumor cells are under hypoxic conditions, it is necessary for them to vascularize to survive and proliferate. Under hypoxic conditions, tumor cells induce HIFs (the activated form of HIF) to promote synthesis and expression of pro-angiogenic factors, such as VEGF and EPO (erythropoietin) [62]. Recent studies reveal that HIF and UPR cooperate to regulate the level of VEGF and the activity of angiogenesis [63]. However, even if VEGF is secreted, it will be sequestered by ECM. Therefore, macrophages will be recruited and release MMP (Matrix metalloproteinases), such as MMP-9 and MMP-2, which can degrade ECM [64]. After ECM is degraded, VEGF is available for binding with VEGF receptors on vascular endothelial cells. Hence, it not only induces MAPK pathways for endothelial movement but also triggers more proangiogenic factors for the positive effect of angiogenesis [65].

### 5.2. Mechanism of Downstream UPR Regulating Angiogenesis

Insufficient vascular supply causes hypoxia, nutrient deprivation, and consequently a lowered ATP production. These factors are also associated with misfolded proteins, and ER stress and UPR will be triggered. Interestingly, UPR can induce both pro-angiogenesis and anti-angiogenesis.

As for pro-angiogenesis, the transcription factors from the arms of UPR (XBP1s and ATF4) can bind with the VEGF promoter and activate it [66]. According to the experiment of Iwawaki, IRE-1α knockout animals manifested that the reduced VEGF level in placenta accounts for the mortality of models [67]. ATF4 transcriptionally modulate IL-8 (interleukin-8) in various human aortic endothelial cell lines. IL-8 [68], a pro-angiogenic cytokine, can induce endothelial cell proliferation and vessel formation and can promote MMP-2 and MMP-9 expression, which decreases endothelial cell apoptosis [69]. XBP-1s can bind to the VEGF promoter and drive endothelial proliferation through the Akt/GSK/β-catenin axis pathway [70]. XBP1s has been shown to induce triple-negative breast cancer cell proliferation through a combination with HIF-1α to up-regulate the expression of VEGF [71].

Furthermore, activated Nrf2 (NF-E2-related nuclear factors) can cooperate with UPR to trigger a pro-angiogenic response under oxidative stress. Then, HIF-α will be stabilized through UPR signaling, which promotes VEGF expression to make tumor cells deal with hypoxic conditions. By contrast, as for anti-angiogenesis, there are several known mechanisms mentioned below: The activation of TRAF2-JNK by IRE-1α induces cytochrome c-mediated apoptotic pathways through phosphorylating specific Bcl-2 family proteins, such as Bim [72]. Besides, the activation of the PERK-ATF4 axis triggers the expression of CHOP, which enhances the synthesis of apoptotic proteins. Moreover, CHOP inhibits eNOS promoter (endothelial nitric oxide synthase), a significant factor inducing angiogenesis. An example of CHOP negative regulation of vascularization is illustrated in CHOP^−/−^ mice, which shows ischemia-induced neovascularization.

Evidence also shows that a reduction in apoptosis and increased expression of eNOS were observed in the CHOP10 (C/EBP homologous protein-10) knockout mice model compared with wild-type mice [73]. Upon ER stress, CREB3L1 (cyclic AMP (cAMP)-responsive element-binding protein 3-like protein 1), a member of UPR, plays an opposite role on the angiogenesis of highly metastatic breast cancer cells (LN4D6) and its angiogenesis [74]. Under hypoxia, CREB3L1 translocates from ER to the Golgi body, being cleaved by regulated RIP (intermembrane proteolysis) to convert into activated transcription factor. The transcription factor can bind with PTN (Pleiotrophin) or FGFBP1 (Fibroblast growth factor-binding protein 1) promoter, which are crucial for tumor growth and then inactivates them [75]. Recent studies uncovered that metastatic cells transfected with CREB3L1 manifested loss of angiogenesis and failure to migration in vitro [74]. This leads to the conclusion that CHOP and CREB3L1 induced by ER stress play vital roles in the anti-angiogenic response (Figure 5).

## 6. UPR and Metastasis

### 6.1. The Process of Metastasis

Metastasis is cancer cells proliferating from the original colony to a distant site [76]. It results in poorer prognosis and more resistance to therapy [77]. Metastasis is a multistage process, which primarily consists of five steps [76]. In the first step, invasion, cancer cells must pass through the basal membrane to the extracellular matrix [78]. The second step is intravasation, where cancer cells enter the blood or lymphatic vessels for further localization [76,79]. In the third step, dissemination, cancer cells travel to distant sites, where they may seed new metastatic colonies [80]. The fourth step is extravasation, where circulating cancer cells translocate from vessels to secondary destination [76]. The final step is colonization, where metastatic cells adapt to the secondary organ and establish micrometastases or macrometastases [76]. Although cancer cells are greatly successful in metastasizing in the long run, at the cellular level, only 0.02% of metastatic cells can survive and form macrometastases [81].

### 6.2. The Relationship between Metastasis and UPR

We mentioned that there are three integral ER membrane proteins in the UPR, including IRE-1, PERK, and ATF6. However, PERK is most significant for stimulating metastasis because it can maintain endothelial cell survival and promote angiogenesis by VEGF. Epithelial-to-mesenchymal transition (EMT) is a cell transdifferentiation program usually used by cancer cells to migrate and invade. Cancer cells that have undergone an EMT tend to employ the PERK-ATF4 branch of UPR for metastasis [82]. Furthermore, PERK-ATF4 plays a critical role in mediating the expression of pro-invasion EMT-signature genes, especially CREB3L1, an ER-associated transcription factor [82]. CREB3L1 is located on the ER membrane, and its activation is triggered by cleavage through site1 proteases (SP1) and site2 proteases (SP2) [83]. CREB3L1 not only promotes metastasis but also rescues the decrease of invasion upon lack of ATF4. In breast cancer, CREB3L1 is greatly up-regulated, which leads to enhanced metastasis and poor prognosis. Thus, the inhibition of CREB3L1 greatly decreases the activity of FAK—a kinase regulated by ECM interactions and known to be crucial for the migration of cancer cells [82].

## 7. Promising Therapies that Inhibit Cancer Using Mediating UPR

We summarize the anti-cancer drugs targeting on UPR proteins and their mechanisms (Table 1). The categories of the table are arranged according to the order of UPR signaling pathways. The proteins associated with initiation, for instance, Grp78 (Bip), is introduced firstly. Then, it moves to introduce the downstream signaling, such as ATF6, IRE-1, PERK and eIF2α. Lastly, ERAD and chaperone are introduced.

### 7.1. Target for Grp78/Bip

Chaperone Grp78/Bip plays a significant role in cancer cell survival, metastasis, and drug resistance. An example is provided by cisplatin, an anti-cancer agent that interferes with DNA replication to suppress tumor cell proliferation [84]. However, an increased level of Grp78/Bip is attributed to cisplatin-resistance in ovarian cancer. Furthermore, the knockdown of Grp78/Bip promotes the sensitivity of ovarian cancer cell lines to cisplatin [85]. Therefore, through the inhibition of Grp78/Bip, it is possible for cancer cells to enhance their sensitivity to chemotherapy [108]. There are several Grp78/Bip inhibitors, such as 4-PBA (4-Phenylbutyric acid), TUDC (Tauroursodeoxycholic acid) and EGF-SubA (Epidermal Growth Factor-SubA), accessible to suppress tumor cell growth. It is proven that 4-PBA and TUDC can promote cytotoxicity and apoptosis by decreasing the activity of ER stress in human lung cancer A549 and H460 cells combined with cisplatin [86]. Furthermore, it is revealed that EGF-SubA specifically induces cleavage of Grp78/Bip at a di-leucine motif and becomes highly cytotoxic to cancer cells [87]. EGF-SubA can suppress the human breast and prostate tumor xenografts in a mouse model [109]. Additionally, it can also promote the sensitivity of cancer cells to thapsigargin and, therefore, reduce its chemoresistance [109]. EGF-SubA is accessible to promote eIF2α phosphorylation and upregulate the expression of CHOP for apoptosis [11].

Another example is Epigallocatechin-3-gallate (EGCG), which binds with the ATP-binding domain of Grp78 in glioma cells, strengthening the sensitivity of cancer cells to temozolomide or etoposide [10]. Accumulation of Grp78 in the ER promotes the transformation of constitutive UPR of therapy-recalcitrant malignant mesothelioma cells into pro-apoptotic ER stress [110]. Moreover, recent studies show that EGCG serves as a PARP16 inhibitor to promote ER stress-induced cancer cell apoptosis [111]. However, overexpression of Grp78/Bip can inactivate caspase 7, which induces cancer cell progression. A study suggests that EGCG impedes the formation of the GRP78/caspase 7 complex and prevents the anti-apoptotic effects of GRP78 [112]. Nevertheless, EGCG can also interact with other proteins, such as MMP-2, MMP-9, and BCl-2. It can decrease the expression of MMP-2 and MMP-9 through suppression of HGF-Met signaling and triggers apoptosis through downregulating Bcl-2 or upregulating Bax in prostate carcinoma cells, hepatoma cells, bladder carcinoma cells, and ovarian carcinoma cells [113].

Focusing on Grp78-associated angiogenesis or tumor resistance, Grp78 is highly expressed on the surface of proliferative endothelial cells, with the assistance of T-cadherin, and thus endothelial cells can maintain survival [114]. Besides, under hypoxia or the presence of VEGF, endothelial cells are promoted to migrate and proliferate. For example, GRP78-specific mouse monoclonal IgG antibody (mAb159) successfully prevents tumor cell growth and their angiogenesis through inhibiting PI3K signaling [89]. Secretion of Grp78 through certain tumor cell lines makes cancer cells also resistant to pro-apoptotic effects, triggered by Bortezomib. Hence, several drugs, such as camptothecin-11 (CPT-11), Etoposide and Temozolomide, assist in overcoming Grp78-associated tumor resistance in glioma cells [88].

Another promising therapy targeting Grp78/Bip is PAT-SM6, a monoclonal IgM antibody [90]. It recognizes tumor cells, especially human multiple myeloma (MM) cells, expressing Grp78 and inducing complement dependent cytotoxicity (CDC) and antibody-dependent cellular cytotoxicity (ADCC), leading to apoptosis [91]. Furthermore, Vorinostat, an HDAC inhibitor, blocks the histone deacetylase and acetylates Grp78/Bip, which in turn intrudes the function of Grp78/Bip and protein modification. Therefore, it can activate UPR and lead to anti-cancer effects [92].

### 7.2. Target for ATF6α

ATF6α, which upregulates the expression of foldases and chaperones, is activated by proteolysis and translocates from the ER to the Golgi body. Besides, the activation of ATF6α is crucial for cancer cell survival [94]. ATF6α contains disulfide bonds in its luminal domain, and protein disulfide isomerase (PDI) is required for the formation of disulfide bonds [11]. Nevertheless, ATF6α targeting therapies can fall into two categories, including direct and indirect inhibition [11]. Direct inhibitors, ceapins, a class of pyrazole amides, can selectively block ATF6α and trap it in the ER to suppress signal transduction [93,94]. Therefore, ceapins, a class of pyrazole amides are a promising group of small molecules targeting ATF6α to inhibit cancer cell proliferation.

Moreover, indirect inhibitors that mainly target ATF6α-associated enzymes, such as PDI, S1P, and S2P. 4-(2-aminoethyl) benzene-sulfonyl fluoride (AEBSP), a serine protease inhibitor, inhibits the proteases S1P and S2P in the Golgi body, which leads to the blockade of ATF6α downstream signaling pathway [95]. Another drug is propynoic acid carbamoyl methyl amide 31 (PACMA31), a PDI inhibitor, interfering with the stability of ATF6α and causing the failure of ATF6α signal transduction. A study indicates that PDIA5/ATF6 upregulation in leukemia cells confers resistance to Imatinib, a tyrosine kinase inhibitor, and a clinical anticancer drug Glivec^®^). However, 16F16, also a PDI inhibitor, triggers a molecular mechanism similar to that of PACMA31, as well as reducing the chemo-resistance and promoting the sensitivity of cancer cells to Imatinib [96].

### 7.3. Target for IRE-1α

As mentioned in the Introduction, IRE-1α downstream signaling can either promote adaptive signals via XBP1 mRNA splicing or trigger cell death signals via RIDD. Therefore, promising therapies targeting IRE-1 can be mainly classified into two categories. One stimulates RIDD for apoptosis, and the other focuses on blocking XBP1 mRNA splicing. Moreover, the therapies related to XBP1 mRNA splicing can fall into two classes, including ATP binding kinase domain inhibitors and IRE-1 RNase inhibitors [115]. ATP binding kinase domain inhibitors, such as APY29, Sunitinib, and Quercetin block IRE-1α from phosphorylation and then suppress its activation [13].

In contrast, IRE-1 endoribonuclease inhibitors, such as MKC-3946 and 8-formyl-7-hydroxy-4-methylcoumarin (4μ8C), protect the formation of XBP1s (active XBP1) through interacting with the catalytic core of the RNase domain of IRE-1α [13,97]. Quercetin stabilizes the inactive conformation of the ATP-binding site within the kinase domain and indirectly inhibits the activity of its RNase [11]. MKC-3946 does not induce strong apoptosis in human multiple myeloma, whereas it can trigger stronger apoptosis in tumor cells under hypoxic conditions. MKC-3946-driven tumor cell apoptosis is using blocking XBP1 mRNA splicing and increasing the expression of CHOP [11]. A small-molecule inhibitor of the endoribonuclease (RNase) activity of IRE-1α, 4μ8c, can block the pathway to suppress the proliferation of H4IIE hepatoma cells [116].

### 7.4. Target for PERK

As mentioned above, PERK can not only activate its downstream transcription factor, ATF4, to increase cancer cell survival but can also trigger Nrf2, also a transcription factor, to block expression of CHOP, which induces cell death signaling. Thus, GSK2656157, a small molecule PERK inhibitor, possesses ATP-competitive activity to interact with elF2α and inactivates it (phosphorylated elF2α) [98]. Furthermore, a study reveals that PERK-deficient MEFs manifest decreased cell growth and increased apoptosis under hypoxic micro-environments, confirming that PERK inhibitors have anti-cancer activity [117]. Another experiment shows that PERK inhibitors attenuate cancer cell growth in pancreatic adenocarcinoma [118]. Besides, GSK2656157 can also block the activity of PERK to suppress angiogenesis [119]. Unfortunately, inhibition of PERK by GSK2656157 leads to several side effects, such as harming pancreatic exocrine and beta cells [118].

### 7.5. Target for eIF2α

Two synthetic compounds, Salubrinal and Guanabenz, target the GADD34/PP1c complex and protect eIF2α from dephosphorylation, which stimulates eIF2α phosphorylation, induces expression of CHOP and triggers apoptosis [99,100]. Evidence suggests that treating hepatoma cells with Salubrinal induces eIF2α phosphorylation and the expression of CHOP, and enhances caspase activation for apoptosis [120]. Another study shows that treating MCF-7/ADR (acquired Doxorubicin resistance) breast cancer cells, which are highly resistant to Doxorubicin, with Salubrinal can promote doxorubicin-mediated apoptosis by blocking GADD34 and increasing eIF2 phosphorylation, resulting in the inhibition of translation of eIF2α-dependent proteins [121].

### 7.6. Target for ERAD

ERAD, a member of the proteasome, is accessible to eliminate the accumulation of misfolded proteins. Bortezomib, an ERAD inhibitor, is widely used in multiple myeloma and mantle cell lymphoma for retarding the proliferation of cancer cells [101]. It inhibits ERAD and suppresses cancer cell proliferation through specifically blocking the 26S proteasome and intruding its proteolysis [102]. Another ERAD inhibitor, Ritonavir, is similar to Bortezomib, and it can cripple the ERAD system, causing misfolded or unfolded protein overloading [103]. EerI (Eeyarestatin I), a chemical inhibitor of ERAD, is similar to Bortezomib, but it can also trigger apoptosis through up-regulating the Bcl-2 homology3 (BH3)-only pro-apoptotic protein NOXA [104]. Besides, Eerl can interact with p97 ATPase and block ERAD [104]. Additionally, both Eerl and Bortezomib are not only accessible to bind with the NOXA promoter and then activate it, but also intercept the ubiquitination of histone H2A to induce the transcription of NOXA [101].

### 7.7. Target for Chaperones

The chaperone inhibitor MAL3-101 interferes with the ATPase activity of the HSP70 protein and brings about the accumulation of misfolded protein and apoptosis [105]. For example, the HSP70 Modulator, an allosteric Hsp70 inhibitor, MAL3-101 inhibits Merkel cell carcinoma (MCC), neuroendocrine carcinoma of the skin [105]. Two other therapies targeting chaperones are Retaspimycin (IPI-504) and SNX-2112. They competitively interact with the N-terminal ATP-binding site of HSP90 and lead to the instability of oncogenic kinases, causing cell cycle arrest or apoptosis [106,107]. SNX-2112 mainly induces cell cycle G2/M blocking and intrudes the expression of EGFR (epidermal growth factor receptor) and ERK1 and -2 (extracellular signal-regulated kinases 1 and 2) [122].

## 8. Conclusions

ER stress plays a significant role in both normal and tumor cells. As for cancer development, UPR is a double-edged sword, as it can not only suppress tumor cell proliferation but also induce tumor cell survival, progression, and metastasis under unfavorable conditions, such as hypemic hypoxia, accumulation of reactive oxygen species, and ATP as well as nutrient deprivation. Tumor cells take advantages of the UPR pathway for their survival. For example, they decrease mRNA stabilities of CHOP via PERK and its downstream signals, including ATF4 and Nrf2 for survival. Besides, they can upregulate the mTOR signaling pathway through ATF6, which increases survival. They also trigger cell cycle G0–G1 arrest by IRE-1α and PERK signaling to escape from chemotherapies and adversities.

Moreover, they make use of IRE-1α-XBP1 downstream signaling to suppress the function of dendritic cells and the maturation of the immune system. As for metastasis, tumor cells transfer themselves to distant sites via PERK-ATF4. After metastasis, tumor cells overcome hypoxia and nutrient deprivation through IRE-1α-induced and PERK-driven angiogenesis. With increasing knowledge of the relation between UPR and tumor cells, several anti-cancer therapies will become feasible. This holds particularly for drugs targeting the UPR signaling pathway that combined with other already available anti-cancer therapies can enhance anticancer therapeutic effects while decreasing side effects.

## Figures and Tables

**Figure 1 ijms-20-02518-f001:**
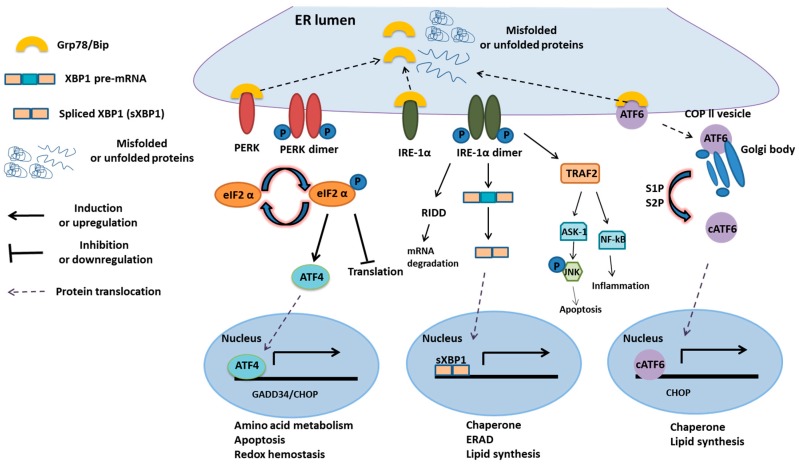
Overview of the UPR process [14]. Upon stimulation of the unfolded protein response, the Grp78/Bip is recruited as a chaperone and leaves its interaction with the three transmembrane proteins of the unfolded protein response (UPR), the IRE-1α, PERK, and ATF6. This allows these proteins to oligomerize and become activated PERK dimerizes and phosphorylates the eIF2α suppressing 5’capped mRNA translation.

**Figure 2 ijms-20-02518-f002:**
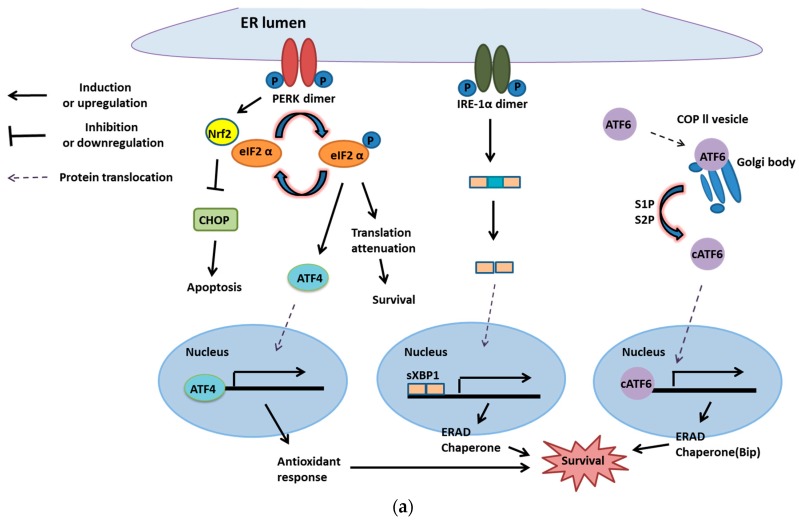
Unfolded protein response and cell survival or death. (**a**) PERK provides cancer cell survival. PERK can activate ATF4, which upregulates the genes with roles in antioxidant response for survival. Moreover, PERK can stimulate Nrf2 to inactivate cell death signaling, CHOP, and phosphorylated elF2α to attenuate translation for survival. Another transmembrane protein of the UPR membrane, IRE-1, and ATF6 also have crucial roles in cancer cell survival. Under the moderate level of ER stress, activated IRE-1 removes the introns of inactivated XBP1 to form spliced XBP1 (XBP1s). XBP1s serves as a transcription factor and binds with the promoter of chaperone and ERAD genes for modifying or degrading misfolded proteins for cell survival. Besides, ATF6 translocates from the ER membrane to the Golgi body. After moving to the Golgi body, ATF6 is cleaved to release the transcription factor (active segment) that induces the expression of chaperones and ERAD [42]; (**b**) When cells are overloaded with misfolded proteins, three transmembrane proteins of UPR are inclined to trigger cell death signals. Activated PERK phosphorylates elF2 to block protein synthesis. Furthermore, inactive elF2 will induce ATF4, a transcription factor that promotes Noxa and CHOP (both are pro-apoptotic transcription factors). Then, CHOP stimulates Bim, a pro-apoptotic protein of Bcl-2 families, and directly activates Bax and Bak on the membrane of mitochondria to trigger apoptosis. Furthermore, once IRE-1 is phosphorylated by extensive UPR, it will recruit TRAF 2 and activate apoptosis signal-regulating kinase 1 (ASK1) to phosphorylate JNK. Activated JNK can inhibit anti-apoptotic proteins, such as Mcl-1 and Bcl-XL, to trigger cell death signaling. Another pathway, cleaved ATF6, also induces CHOP expression and leads to apoptosis.

**Figure 3 ijms-20-02518-f003:**
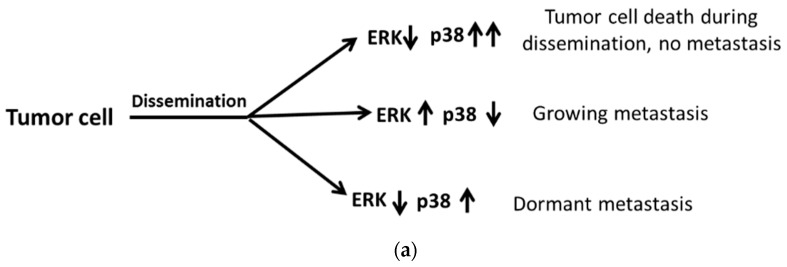
The role of dormancy in metastasis and chemoresistance of cancer cells. (**a**) Dormancy-regulated metastasis affected by the balance between p38 and ERK. When tumor cells disseminate, there are three different outcomes in different microenvironments. Firstly, with upregulation of MAPK p38 combined with decreasing ERK signals, tumor cells degenerate without metastasis during dissemination; Secondly, increasing ERK with decreasing MAPK p38 promotes metastasis; Thirdly, decreasing ERK with increasing p38 brings about metastatic cells dormancy; (**b**) UPR-induced dormancy is associated with cancer cell survival and chemoresistance. MAPK p38 can suppress FoxM1, c-Jun and the uPAR (Urokinase-type plasminogen activator receptor) transcript, which is crucial for the activation of ERK. Furthermore, it can also trigger the downstream signaling of UPR. Activated PERK can phosphorylate elF2α for the G0–G1 arrest and induce ATF4 for survival. Besides, phosphorylated elF2α can trigger ATF4 for survival. Activated IRE-1α can induce Grp78/Bip for survival as well as block the pro-apoptotic signal, Bax. Furthermore, ATF6α can promote survival through the mTOR signaling pathway. Redrawn from Sosa et al. [48].

**Figure 4 ijms-20-02518-f004:**
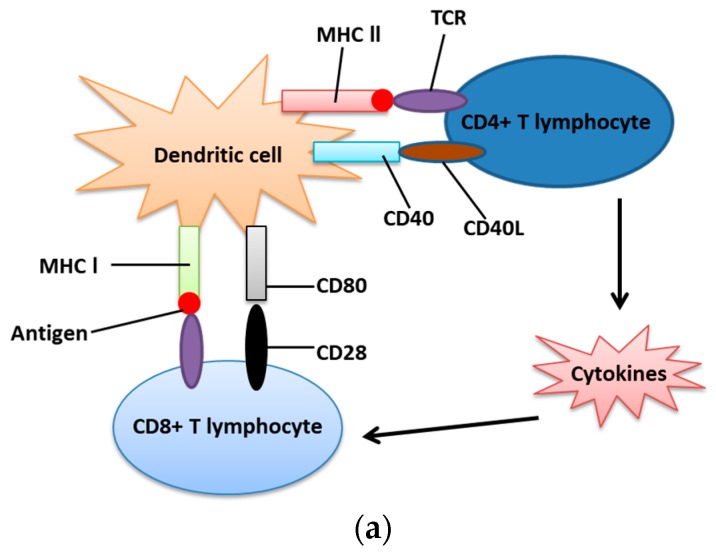
Immunocompetence and immunosuppression. (**a**) Immunocompetence. Dendritic cells (DCs), known as antigen presenting cells, use its CD40 to bind with CD40L (CD40 ligand) on the surface of CD4+ T lymphocyte. Furthermore, DC also uses its MHC-II, combined with antigen, to bind with TCR on CD4+ T lymphocyte. Therefore, CD4+ T lymphocyte is activated by DC to secrete cytokines or to activate B cells. DC can also activate CD8+ T lymphocyte via two signal transductions. One is CD80 and CD28 interaction, and the other is MHC-I and TCR combination; (**b**) Immunosuppression. When the tumor dendritic cell is under stress, such as hypoxia, nutrient deprivation and accumulation of ROS, it activates 4-NHE to induce IRE1-α and its downstream transcription factor, XBP1. This effect leads to the failure of antigen presentation and, therefore, is not able to activate T lymphocytes. Besides, activated XBP1 (sXBP1) can increase the production of phosphatidylcholine, which inhibits the ability of antigen presentation. Redrawn from Cubillos-Ruiz et al. [32].

**Figure 5 ijms-20-02518-f005:**
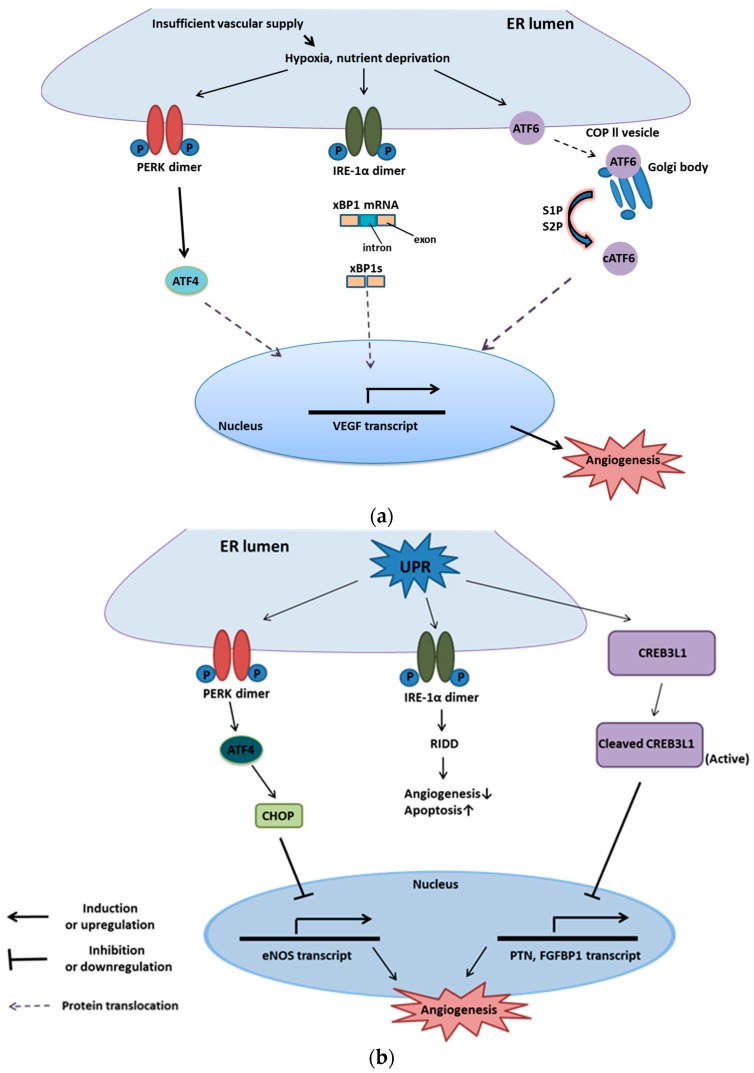
UPR and angiogenesis. [75] (**a**) UPR and pro-angiogenesis. Several low-stress adversities, such as hypoxia and nutrient deprivation, are accessible to activate three branches of UPR (IRE-1, PERK, and ATF6) on the ER membrane. Active IRE-1, PERK, and ATF6 can trigger transcription factors, namely XBP1s, ATF4, and cleaved ATF6, respectively, to upregulate VEGF expression, which leads to endothelial cell survival, proliferation and migration for angiogenesis; (**b**) UPR and anti-angiogenesis. Severe hypoxia and nutrient deprivation stimulate UPR to activate PERK on the ER membrane. Activated PERK can induce ATF4, a transcription factor, to upregulate the expression of CHOP. CHOP also serves as a transcription factor and is accessible to bind with pro-angiogenic mRNA, such as eNOS, to inhibit angiogenesis. Furthermore, high-intensity ER stresses trigger UPR and stimulate CRE3BL1 to be cleaved by RIP (regulated intermembrane proteolysis) to generate active CRE3BL1. Then, the activated CRE3BL1 translocates to the nucleus and binds with pro-angiogenic mRNA, such as PTN (Pleiotrophin) and FGFBP1 (Fibroblast growth factor-binding protein1), to block angiogenesis.

**Table 1 ijms-20-02518-t001:** Anti-cancer drugs targeting UPR proteins.

Stage	Target	Mechanism		Small Molecule	Outcome	Reference
Initiation	Grp78 (Bip)	Unknown		4-Phenyl-butyric acid (4-PBA); Tauroursodeoxycholic acid (TUDC)	Blocking ER stress to induce cytotoxicity and apoptosis	[84,85,86]
		Specifically cleaving Grp78 at a di-leucine motif		EGF-SubA	Leading to high cytotoxicity and reducing chemo-resistance	[11,87]
		Binding to the ATP-binding site of Grp78 and modulating its ATPase activity		Epigallocatechin-3-gallate (EGCG)	Enhancing ER stress-induced cancer cell apoptosis	[10]
		Inhibiting the ATPase activity of Grp78		CPT-11, etoposide, and temozolomide	Increasing sensitivity of cancer cells to bortezomib	[88]
		Triggering Grp78 endocytosis		Mouse monoclonal antibody (mAb159)	Inhibiting endothelial cells and angiogenesis	[89]
		Recognizing tumor cells expressing Grp78 and inducing complement dependent cytotoxicity (CDC) and antibody-dependent cellular cytotoxicity (ADCC)		PAT-SM6 (monoclonal IgM)	Inducing MM cell death	[90,91]
		Blocking the histone deacetylase and bringing about acetylation of Grp78		Vorinostat (HDAC Inhibitors)	Intruding the function of Grp78, which contributes to the accumulation of misfolded protein and cell death	[92]
Sensor	ATF6	Directly target ATF6	Selectively blocking ATF6 and trapping it in the ER	Ceapins	Sensitizing cancer cells to ER stress	[93,94]
		Indirectly target ATF6, mainly target associated enzymes	Hindering the proteases S1P and S2P in the Golgi body	4-(2-Aminoethyl) benzene- sulfonyl fluoride (AEBSP)	Blocking nuclear localization and inhibiting ATF6 downstream signaling	[95]
			Unknown	Propynoic acid carbamoyl methyl amide 31 (PACMA31)	Block ATF6 downstream signaling	[96]
			Inhibit the disulfide bond formation of ATF6	16 F16	Reducing the chemo-resistance and promoting sensitivity to Imatinib	[96]
Sensor	IRE-1-XBP1	Interacting with the catalytic core of the RNase domain of IRE-1α		8-Formyl-7-hydroxy-4-methyl coumarin (4μ8c)	Inhibiting the endoribonuclease (RNase) activity of IRE1	[11,13,97]
				MKC-3946	Inhibiting RNase activity of IRE-1 and increase expression of CHOP	[11,13,97]
		Binding with the ATP binding site within the IRE-1 kinase domain		APY29, Sunitinib	Inhibiting IRE-1 phosphorylation and indirectly suppressing its RNase activity	[11,13]
		Stabilizing the inactive conformation of the ATP-binding site within the IRE-1 kinase domain		Quercetin		
	PERK	Competing with the ATP-binding site within PERK kinase		GSK2656157	Inhibiting PERK autophosphorylation and phosphorylation of eIF2α	[98]
Downstream	eIF2α	Interrupting the activity of GADD34/PP1c complex and protecting eIF2α from dephosphorylation		Salubrinal; Guanabenz	Stimulating eIF2α phosphorylation, inducing expression of CHOP and trigger apoptosis	[99,100]
ERAD	ERAD	Blocking the 26S proteasome and intrude proteolysis		Bortezomib	Inhibiting ERAD and retarding the proliferation of cancer cells	[101,102]
				Ritonavir	Crippling the ERAD system, and causing misfolded protein overloading	[102,103]
		Interacting with p97 ATPase and block ERAD		Eeyarestatin I	Triggering NOXA and inducing cancer cell apoptosis	[104]
Chaperone	HSP70	Interfering with the ATPase activity of HSP70 proteins		MAL3-101	Blocking the function of HSP70, which leads to the accumulation of misfolded protein and apoptosis	[105]
	HSP90	Competitively interacting with the N-terminal ATP-binding site of HSP90		Retaspimycin (IPI-504)	Leading to the instability of oncogenic kinases and bringing about cell cycle arrest or apoptosis	[106,107]
				SNX-2112		

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
