# Peer review of "Unfolded Protein Response (UPR) in Survival, Dormancy, Immunosuppression, Metastasis, and Treatments of Cancer Cells"

_ijms, 2019, doi:10.3390/ijms20102518_

Round 1

Reviewer 1 Report

I have now checked the manuscript revised by the authors. The article can now be accepted in its present form.

Reviewer 2 Report

Although a resubmission, this manuscript has not changed significantly from its previous submission. A review is meant to clear up and enlighten but this manuscript remains confusing and jumbled.  Other than a limited introduction to the UPR components, the authors did not improve the text sufficiently to allow the reader to gain the intended insight.  The first confusion that was encountered was reference to the ' three integral ER membrane proteins' that are components of the UPR as if they are the only integral membrane proteins in the ER.  

Rather than the convoluted text of the authors, the functions of these proteins can be stated very simply:  when BIP is bound to the UPR membrane elements, they are inactive. When BIP is recruited as a protein chaperone by the presence of a large amount of unfolded or misfolded protein, then the UPR membrane elements multimerize and become activated.  Upon activation, the PERK phosphorylates....., the IRE-1..........

As another example, here is a rewrite of the legend for Figure 1. 

Figure 1. Overview of the UPR process. Upon stimulation of the unfolded protein response, the Grp78/Bip is recruited as a chaperone and leaves its interaction with the  three transmembrane proteins of the unfolded protein response (UPR),  the IRE-1α, PERK, and ATF6.  This allows these proteins to oligomerize and become activated  PERK dimerizes and   phosphorylates the eIF2α suppressing  5’capped mRNA translation **editors note NOT ALL PROTEIN SYNTHESIS IS Eif2a DEPENDENT** . The activated IRE-1α uses its endonuclease activity to cleave and resplice  XBP1 mRNA to create potent transcription factor to upregulate the expression of UPR-related genes, including chaperones, lipid synthesis, and ERAD. The activated IRE-1α also cleaves other RNAs in the ER to promote their decay in a process termed regulated IRE1-dependent decay (RIDD).   Furthermore, the IRE-1α  oligomer can also interact with TRAF2 to trigger the JNK pathway for cell apoptosis.  The Bip-free ATF6 translocates from the ER to the Golgi body by a COPII vesicle where it is activated by sequential cleavage  by S1P and S2P proteases. The cleaved ATF6 can enter the nucleus and upregulate the expression of CHOP, chaperones, and lipid synthesis.

Similarly, the rest of the text requires extensive rewriting to allow the reader to understand the message.  The entire text must be rewritten in a simpler, more organized, logical,  step by step manner before it can be acceptable. 

Reviewer 3 Report

The review is well written, timely and really informative for cancer treatment and especially the role of of Unfolded Protein Response in tumor progression, I recommend it for publication.

This manuscript is a resubmission of an earlier submission. The following is a list of the peer review reports and author responses from that submission.

Round 1

Reviewer 1 Report

I have now completed reviewing the manuscript " The role of UPR in cell survival, dormancy immunosuppression, angiogenesis, metastasis and treatments of cancer cells".

The article has numerous typos in the body which needs attention.

The figure formats are not acceptable and not in a publishable manner

There are sections in the manuscript like "The process of metastasis"  which completely lacks referencing.

When discussing about negative implications of UPR in cancer, authors need to mention about Grp78 and its role in cell migration, invasion and role in metastasis which is very important. In reality, there needs to be a section on how UPR proteins influence cancer progression denoting individually about UPR proteins and their mechanism and role in cancer.

There are a lot of pre-existing content in the review which are not novel and already published which is unnecessarily increasing the bulk of the review and making it lengthy.

Overall, I believe, authors need to re-modify and refurbish the review with regards to non technical and technical aspects and later re-submit it.

Reviewer 2 Report

The review by Sheng-Kai Hsu et al on the role of the unfolded protein response (UPR) on cancer   attempts to provide an extensive overview of all possible interactions of the UPR and its components with all aspects of cancer from growth, angiogenesis, metastasis to immune control and the potential to enhance chemotherapy using inhibitors of the UPR.  A useful table of inhibitors of the UPR and several figures are provided. The figures provide a reasonable explanation of the relevant biology and the figure legends summarize the text but still not written well enough. The pictures need to be redone using  a drawing program, at least using powerpoint, rather than sketches that are hard to read in places.  The table  of inhibitors and the text are a useful collection but could be more integrated into the manuscript as each element of the UPR is introduced with the consequence of inhibition on cancer cell function and potential for cancer therapy. 

The manuscript is very difficult to read. It is repetitive, the descriptions are inaccurate and convoluted.  The English word usage and logical expression is sufficiently poor that it is hard to follow.  Most significantly, the authors fail to give a cohesive, concise, understandable description of the UPR and instead, they attempt to explain the peripheral actions of the UPR on angiogenesis, metastasis and immune control.  The connections are less clear and the discussion distracts from the main theme. The manuscript needs to be rewritten in a much shorter focused version.  Normally I provide a list of suggested changes to improve the manuscript but unfortunately, there are too many to do so for this review.  The following are the more significant issues:

Legend to Figures are inaccurate and hard to follow but more succinct than the text.

The usage of many English words is incorrect or inaccurate: e.g. protect, intruding, highly, etc.

There is no mention of induction of UPR by thapsigargin in the beginning but it is referred to later.

No mention that Grp78 normally binds to UPR components preventing their activation.   Simply stated,  in the presence of excess unfolded proteins, Grp78 moves from its interaction with PERK, ATF6 and IRE-1a to act as a chaperone. This allows activation of ATF6 and PERK and IRE-1a to dimerize and become active.

Poor discussion of IRE-1a and its RNAse activity. Line 100: Ire-1a is NOT “a transmembrane lipid.”

Poor discussion of GADD34 and phosphatase

Section 1.1 has a very broad title, ‘functions of the ER’ and yet this section is very incomplete.

Section starting Line 220: The concepts of dormancy were very confusing and poorly written. It could be simplified.

Section starting 290. Immunology: This section was not relevant to the manuscript and had inaccuracies.  CD40L is on the activated T cell and not as indicated.  The CD40L-CD40 story was inaccurate. This inaccuracy puts the accuracy of other aspects of the manuscript in question. 

The immunology section was so disjointed that it was hard to see the relevance.

Section starting 348.  The angiogenesis story jumped around with much too much introduction.

Section starting 431 Metastasis story jumped around and was hard to follow.

More care needs to be taken in the descriptions of the inhibitors.